# Super-Acceleration with Cyclical Step-sizes

## Abstract

Cyclical step-sizes are becoming increasingly popular in the optimization of deep learning problems. Motivated by recent observations on the spectral gaps of Hessians in machine learning, we show that these step-size schedules offer a simple way to exploit them. More precisely, we develop a convergence rate analysis for quadratic objectives that provides optimal parameters and shows that cyclical learning rates can improve upon traditional lower complexity bounds. We further propose a systematic approach to design optimal first order methods for quadratic minimization with a given spectral structure. Finally, we provide a local convergence rate analysis beyond quadratic minimization for the proposed methods and illustrate our findings through benchmarks on least squares and logistic regression problems.

## 1   Introduction

One of the most iconic methods in first order optimization is gradient descent with momentum, also known as the heavy ball method [Polyak, 1964]. This method enjoys widespread popularity both in its original formulation and in a stochastic variant that replaces the gradient by a stochastic estimate, a method that is behind many of the recent breakthroughs in deep learning [Sutskever et al., 2013].

A variant of the stochastic heavy ball where the step-sizes are chosen in *cyclical* order has recently come to the forefront of machine learning research, showing state-of-the art results on different deep learning benchmarks [Loshchilov and Hutter, 2017, Smith, 2017]. Inspired by this empirical success, we aim to study the convergence of the heavy ball algorithm where step-sizes $h_0, h_1, \ldots$ are not fixed or decreasing but instead chosen in cyclical order:

---

**Algorithm 1:** Cyclical heavy ball $\mathrm{HB}_K(h_0, \ldots, h_{K-1}; m)$

**Input:** Initialization $x_0$, momentum $m \in (0, 1)$, step-sizes $\{h_0, \ldots, h_{K-1}\}$

$x_1 = x_0 - \dfrac{h_0}{1+m}\nabla f(x_0)$

**for** $t = 1, 2, \ldots$ **do**      $x_{t+1} = x_t - h_{\mathrm{mod}(t,K)}\nabla f(x_t) + m(x_t - x_{t-1})$

**end**

---

The heavy ball method with constant step-sizes enjoys a mature theory, where it is known for example to achieve optimal black-box worst-case complexity of quadratic convex optimization [Nemirovsky, 1992]. In stark contrast, little is known about the the convergence of the above variant with cyclical step-sizes. Our main motivating question is

> Do cyclical step-sizes improve convergence of heavy ball?

Submitted to 35th Conference on Neural Information Processing Systems (NeurIPS 2021). Do not distribute.

Our **main contribution** provides a positive answer to this question and, more importantly, *quantifies* the speedup under different assumptions. In particular, we show that for quadratic problems, whenever Hessian's spectrum belongs to two or more disjoint intervals, the heavy ball method with cyclical step-sizes achieves a faster worst-case convergence rate. Recent works have shown that this assumption on the spectrum is quite natural and occurs in many machine learning problems, including deep neural networks [Sagun et al., 2017, Papyan, 2018, Ghorbani et al., 2019, Papyan, 2019]. More precisely, we list our main contributions below.

- In sections 3 and 4, we provide a **tight convergence rate analysis** of the cyclical heavy ball method (Theorems 3.1 and 3.2 for two step-sizes, and Theorem 4.8 for the general case). This analysis highlights a regime under which this method achieves a faster worst-case rate than the accelerated rate of heavy ball, a phenomenon we refer to as *super-acceleration*. Theorem 5.1 extends the (local) convergence rate analysis results to non-quadratic objectives.

- As a byproduct of the convergence-rate analysis, we obtain an explicit expression for the **optimal parameters** in in the case of cycles of length two (Algorithm 2) and an implicit expression in terms of a system of $K$ equations in the general case.

- Section 6 presents **numerical benchmarks** illustrating the improved convergence of the cyclical approach on 4 problems involving quadratic and logistic losses on both synthetic and a handwritten digits recognition dataset.

- Finally, we conclude in Section 7 with a discussion of this work's **limitations**.

## 2 Notation and Problem Setting

Throughout the paper, we consider the problem of minimizing quadratic functions of the form

$$\min_{x \in \mathbb{R}^d} f(x) \,, \quad \text{with} \quad f \in \mathcal{C}_\Lambda \triangleq \left\{ f : f(x) = \tfrac{1}{2}(x - x_*)^T H (x - x_*) + f_*, \ \mathrm{Sp}(H) \subseteq \Lambda \right\} \,, \quad \text{(OPT)}$$

where $\mathcal{C}_\Lambda$ is the class of quadratic functions whose spectrum $\mathrm{Sp}(H)$ is localized in $\Lambda \subseteq [\mu, L] \subseteq \mathbb{R}_{>0}$. We discuss more general settings beyond quadratic minimization in Section 5.

The condition $\Lambda \subseteq [\mu, L]$ implies all quadratic functions under consideration are $L$-smooth and $\mu$-strongly convex. For this function class, we define $\kappa$, the (inverse) condition number, and $\rho$, the ratio between the center of $\Lambda$ and its radius, as

$$\kappa \triangleq \tfrac{\mu}{L}, \qquad \rho \triangleq \tfrac{L+\mu}{L-\mu} \ = \left( \tfrac{1+\kappa}{1-\kappa} \right). \tag{1}$$

Finally, for a method solving (OPT) that generates a sequence of iterates $\{x_t\}$, we define its worst-case rate $r_t$ and its asymptotic rate factor $\tau$ as

$$r_t \triangleq \sup_{x_0 \in \mathbb{R}^d, \ f \in \mathcal{C}_\Lambda} \frac{\|x_t - x_*\|}{\|x_0 - x_*\|}, \qquad 1 - \tau \triangleq \limsup_{t \to \infty} \sqrt[t]{r_t}. \tag{2}$$

## 3 Super-acceleration with Cyclical Step-sizes

---
**Algorithm 2:** Cyclical ($K = 2$) heavy ball with with optimal parameters

---
**Input:** Initialization $x_0$, $\mu_1 < L_1 < \mu_2 < L_2$      (where $L_1 - \mu_1 = L_2 - \mu_2$)

**Set:** $\rho = \frac{L_2 + \mu_1}{L_2 - \mu_1}$, $R = \frac{\mu_2 - L_1}{L_2 - \mu_1}$, $m = \left( \frac{\sqrt{\rho^2 - R^2} - \sqrt{\rho^2 - 1}}{\sqrt{1 - R^2}} \right)^2$

$x_1 = x_0 - \frac{1}{L_1} \nabla f(x_0)$
**for** $t = 1, 2, \ldots$ **do**
     $h_t = \frac{1+m}{L_1}$    (if $t$ is even),      $h_t = \frac{1+m}{\mu_2}$    (if $t$ is odd)
     $x_{t+1} = x_t - h_t \nabla f(x_t) + m(x_t - x_{t-1})$
**end**

---

In this section we develop one of our main contributions, a convergence rate analysis of the cyclical heavy ball method with cycles of length 2. This analysis crucially depends on the location of the Hessian's eigenvalues; we assume that these are contained in a set $\Lambda$ that is the union of 2 intervals *of the same size*

$$\Lambda = [\mu_1, L_1] \cup [\mu_2, L_2], \quad L_1 - \mu_1 = L_2 - \mu_2. \quad (3)$$

By symmetry, this set is alternatively described by

$$\mu \triangleq \mu_1, \quad L \triangleq L_2 \quad \text{and} \quad R \triangleq \frac{\mu_2 - L_1}{L_2 - \mu_1}, \quad (4)$$

where $R$ is the relative length of the gap $\mu_2 - L_1$ with respect to the diameter $L_2 - \mu_1$ (see Figure 1). This parametrization will reveal very convenient as the relative gap will play a crucial role in the convergence rate analysis. Note also that the gap assumption comes without loss of generality, as we allow $R = 0$.

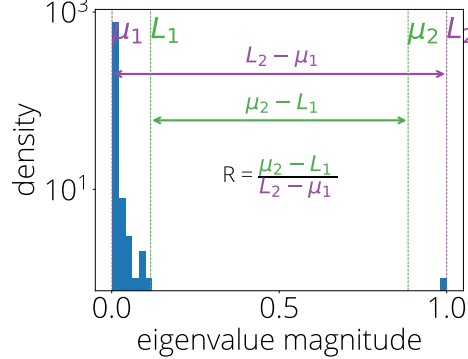

Figure 1: Hessian eigenvalue histogram for a quadratic objective on MNIST. The outlier eigenvalue at $L_2$ generates a non-zero relative gap $R = 0.77$. Under these conditions, the 2-cycle heavy ball method has a faster asymptotic rate than the single-cycle one (see Section 3.1).

Through a correspondence between optimization methods and polynomials that we expand upon in Section 4, we can derive a worst-case analysis for the cyclical heavy ball method. The outcome of this analysis is in the following theorem, that provides the asymptotic convergence rate of Algorithm 1 for cycles of length two. All proofs of results in this section can be found in Appendix D.3.

**Theorem 3.1** (Rate factor of $\mathrm{HB}_2(h_0, h_1; m)$). *Let $f \in \mathcal{C}_\Lambda$ and $h_0, h_1, m \geq 0$. The asymptotic rate factor of Algorithm 1 with cycles of length two is*

$$1 - \tau = \begin{cases} \sqrt{m} & \text{if } \sigma_{\text{sup}} \leq 1, \\ \sqrt{m}\left(\sigma_{\text{sup}} + \sqrt{\sigma_{\text{sup}}^2 - 1}\right)^{\frac{1}{2}} & \text{if } \sigma_{\text{sup}} \in \left(1, \frac{1+m^2}{2m}\right), \\ \geq 1 \text{ (no convergence)} & \text{if } \frac{1+m^2}{2m} \leq \sigma_{\text{sup}}, \end{cases} \quad (5)$$

*with* $\quad \sigma_{\text{sup}} = \sup\limits_{\lambda \in \left\{\mu_1, L_1, \mu_2, L_2, \frac{h_0 + h_1}{2h_0 h_1}\right\} \cap \Lambda} \left| 2\left(\frac{1 + m - \lambda h_0}{2\sqrt{m}}\right)\left(\frac{1 + m - \lambda h_1}{2\sqrt{m}}\right) - 1 \right|. \quad (6)$

This theorem gives the convergence rate for all triplets $(m, h_0, h_1)$. By evaluating this expression over a grid of step-sizes, Figure 2 shows how the rate changes as a function of both step-sizes:

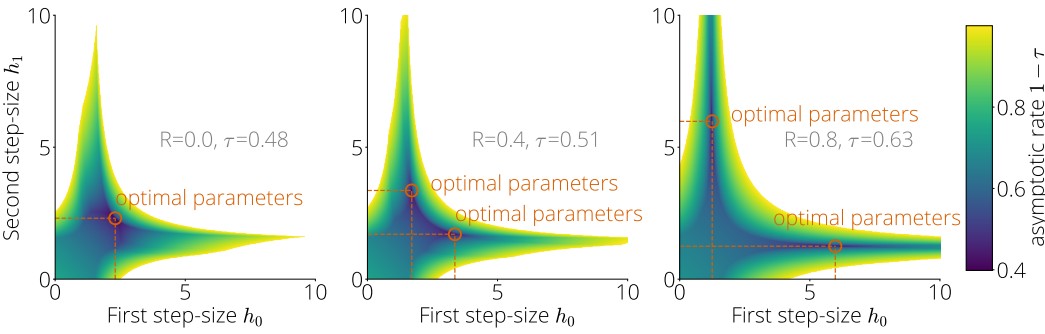

Figure 2: **Asymptotic rate of cyclical** ($K = 2$) **heavy ball** in terms of its step-sizes $h_0, h_1$ across 3 different values of the relative gap $R$. In the **left** plot, the relative gap is zero, and so the step-sizes with smallest rate coincide ($h_0 = h_1$). For non-zero values of $R$ (**center and right**), the optimal method instead alternates between two *different* step-sizes. In all plots the momentum parameter $m$ is set according to Algorithm 2.

From the asymptotic rate expression of Theorem 3.1 we can optimize over the parameters $(h_0, h_1, m)$ to obtain the method with smallest convergence rate. This leads to our other main contribution of this section, the *asymptotically optimal* Algorithm 2. This algorithm enjoys the following rate:

**Corollary 3.2.** *The worst-case (asymptotic) rates $r_t^{Alg.\ 2}$ and $1 - \tau^{Alg.\ 2}$ of Algorithm 2 over $\mathcal{C}_\Lambda$ are*

$$r_t^{Alg.\ 2} = \left(1 + t\sqrt{\frac{\rho^2-1}{\rho^2-R^2}}\right)\left(\frac{\sqrt{\rho^2-R^2}-\sqrt{\rho^2-1}}{\sqrt{1-R^2}}\right)^t, \quad 1 - \tau^{Alg.\ 2} = \frac{\sqrt{\rho^2-R^2}-\sqrt{\rho^2-1}}{\sqrt{1-R^2}} \quad \text{for } t \text{ even.}$$

## 3.1 Comparison with Polyak Heavy Ball

In the absence of eigenvalue gap ($R = 0$ and $\Lambda = [\mu, L]$), Algorithm 2 reduces to Polyak heavy ball (PHB) [Polyak, 1964], whose worst-case rate is detailed in Appendix B. Since the asymptotic rate of Algorithm 2 is monotonically decreasing in $R$, it is always better or equal than PHB. Furthermore, in the ill-conditioned regime (small $\kappa$), the comparison is particularly simple: the optimal 2-cycle algorithm has a $\sqrt{1-R^2}$ relative improvement over PHB, as provided by the next proposition. A more thorough comparison for different support sets $\Lambda$ is discussed in Table 1.

**Proposition 3.3.** *Let $R \in [0, 1)$. The rate factors of respectively Algorithm 2 and PHB verify*

$$1 - \tau^{Alg.\ 2} \underset{\kappa \to 0}{=} 1 - \frac{2\sqrt{\kappa}}{\sqrt{1-R^2}} + o(\sqrt{\kappa}), \qquad 1 - \tau^{PHB} \underset{\kappa \to 0}{=} 1 - 2\sqrt{\kappa} + o(\sqrt{\kappa}). \qquad (7)$$

| Relative gap $R$ | Set $\Lambda$ | Rate factor $\tau$ | Speedup $\tau/\tau^{PHB}$ |
|---|---|---|---|
| $R \in [0, 1)$ | $[\mu, \mu + R(L-\mu)] \cup [L - R(L-\mu), L]$ | $\frac{2\sqrt{\kappa}}{\sqrt{1-R^2}}$ | $(1-R^2)^{-\frac{1}{2}}$ |
| $R = 1 - \sqrt{\kappa}/2$ | $[\mu, \mu + \frac{\sqrt{\mu L}}{4}] \cup [L - \frac{\sqrt{\mu L}}{4}, L]$ | $2\sqrt[4]{\kappa}$ | $\kappa^{-\frac{1}{4}}$ |
| $R = 1 - 2\gamma\kappa$ | $[\mu, (1+\gamma)\mu] \cup [L - \gamma\mu, L]$ | indep. of $\kappa$ | $O(\sqrt{\kappa})$ |

Table 1: Case study of the convergence of Algorithm 2 as a function of $R$, in the regime $\kappa \to 0$. The **first line** corresponds to the regime where $R$ is independent of $\kappa$, and we observe a constant gain w.r.t. PHB. The **second line** considers a setting in which $R$ depends on $\sqrt{\kappa}$, that is, the two intervals in $\Lambda$ are relatively small. The asymptotic rate reads $(1 - 2\sqrt[4]{\kappa})^t$, beating the classical $(1 - 2\sqrt{\kappa})^t$ lower bound, unimprovable when $R = 0$. Finally, in the **third line**, $R$ depends on $\kappa$, the two intervals in $\Lambda$ are so small that the convergence becomes $O(1)$, i.e., is independent of $\kappa$.

## 4 A constructive Approach: Minimax Polynomials

This section presents a generic framework (Algorithm 3) that allows designing optimal momentum and step-size cycles for given sets $\Lambda$ and cycle length $K$.

---

**Algorithm 3:** Optimal momentum method with cyclical step-sizes

---

**Input:** Eigenvalue localization $\Lambda$, cycle length $K$, initialization $x_0$.
**Preprocessing:**

    1. Find the polynomial $\sigma_K^\Lambda$ such that it satisfies (16).

    2. Set step-sizes $\{h_i\}_{i=0,...,K-1}$ and momentum $m$ that satisfy resp. equations (21) and (22).

**Set** $x_1 = x_0 - \dfrac{h_0}{1+m}\nabla f(x_0)$
**for** $t = 1, 2, \ldots$ **do**
    $\quad x_{t+1} = x_t - h_{\mathrm{mod}(t,K)}\nabla f(x_t) + m(x_t - x_{t-1})$
**end**

---

We first recall classical results that link optimal first order methods on quadratics and Chebyshev polynomials. Then, we generalize the approach by showing that optimal methods can be viewed as

combinations of Chebyshev polynomials, and minimax polynomials $\sigma_K^\Lambda$ of degree $K$ over the set $\Lambda$. Finally, we show how to recover the step-size schedule from $\sigma_K^\Lambda$.

## 4.1 First Order Methods on Quadratics and Polynomials

A key property that we will use extensively in the analysis is the following link between first order methods and polynomials (see [Hestenes and Stiefel, 1952]).

**Proposition 4.1.** *Let $f \in \mathcal{C}_\Lambda$. The iterates $x_t$ satisfy*

$$x_{t+1} \in x_0 + \mathrm{span}\{\nabla f(x_0), \ldots, \nabla f(x_t)\}, \tag{8}$$

*where $x_0$ is the initial approximation of $x_*$, if and only if there exists a sequence of polynomials $(P_t)_{t\in\mathbb{N}}$, each of degree at most 1 more than the highest degree of all previous polynomials and $P_0$ of degree 0 (hence the degree of $P_t$ is at most $t$), such that*

$$\forall t \quad x_t - x_* = P_t(H)(x_0 - x_*), \quad P_t(0) = 1. \tag{9}$$

**Example 4.2** (Gradient descent)**.** Consider the gradient descent algorithm with fixed step-size $h$, applied to problem (OPT). Then, after unrolling the update, we have

$$x_{t+1} - x_* = x_t - x_* - h\nabla f(x_t) = x_t - x_* - hH(x_t - x_*) = (I - hH)^{t+1}(x_0 - x_*). \tag{10}$$

In this case, the polynomial associated to gradient descent is $P_t(\lambda) = (1 - h\lambda)^t$.

The above proposition can be used to obtain worst-case rates for first order methods by bounding their associated polynomials. Indeed, using the Cauchy-Schwartz inequality in (9) leads to

$$\|x_t - x_*\| \le \sup_{\lambda \in \Lambda} |P_t(\lambda)| \, \|x_0 - x_*\| \implies r_t = \sup_{\lambda \in \Lambda} |P_t(\lambda)|, \quad \text{where } P(0) = 1. \tag{11}$$

Therefore, finding the algorithm with the fastest worst-case rate can be equivalently framed as the problem of finding the polynomial with smallest value on the eigenvalue support $\Lambda$, subject to the normalization condition $P_t(0) = 1$. Such polynomials are referred to as **minimax**. Throughout the paper, we use this polynomial-based approach to find methods with optimal rates.

An important property of minimax polynomials is their *equioscillation* on $\Lambda$ (see Theorem C.1 and its proof for a formal statement).

**Definition 4.3.** (Equioscillation) A polynomial $P_t$ equioscillates on $\Lambda$ if it verifies $P_t(0) = 1$ and there exist $\lambda_0 < \lambda_1 < \ldots < \lambda_t \in \Lambda$ such that

$$P_t(\lambda_i) = (-1)^i \max_{\lambda \in \Lambda} |P_t(\Lambda)|. \tag{12}$$

**Example 4.4** ($\Lambda$ is an interval)**.** The $t$-th order Chebyshev polynomials of the first kind $T_t$ satisfy the *equioscillation* property on $[-1, 1]$. It follows that minimax polynomials on $\Lambda = [\mu, L]$ can be obtained by composing the Chebyshev polynomial $T_t$ with the linear transformation $\sigma_1^\Lambda$:

$$\frac{T_t\left(\sigma_1^\Lambda(\lambda)\right)}{T_t\left(\sigma_1^\Lambda(0)\right)} = \operatorname*{arg\,min}_{P \in \mathbb{R}_t[X], P(0)=1} \sup_{\lambda \in \Lambda} |P(\lambda)|, \quad \text{with } \sigma_1^\Lambda(\lambda) = \frac{L+\mu}{L-\mu} - \frac{2}{L-\mu}\lambda, \tag{13}$$

where $\sigma_1^\Lambda$ maps the interval $[\mu, L]$ to $[-1, 1]$. The optimization method associated with this minimax polynomial is the Chebyshev semi-iterative method [Flanders and Shortley, 1950, Golub and Varga, 1961] (described also in Appendix B.1). This method achieves the lower complexity bound for smooth strongly convex quadratic minimization, see for instance [Nemirovsky, 1995, Chapter 12] or [Nemirovsky, 1992, Nesterov, 2003].

The next proposition provides the main results in this subsection, which is key for obtaining Algorithm 2. It characterizes the even degree minimax polynomial in the setting of Section 3, that is, when $\Lambda$ is the union of 2 intervals of same size. In this case, the minimax solution is also based on Chebyshev polynomials, but composed with a degree-two polynomial $\sigma_2^\Lambda$.

**Proposition 4.5.** *Let $\Lambda = [\mu_1, L_1] \cup [\mu_2, L_2]$ be an union of two intervals of the same size ($L_1 - \mu_1 = L_2 - \mu_2$) and let $m$ be as defined in Algorithm 2. Then the minimax polynomial (solution to (12)) is, for all $t = 2n$, $n \in \mathbb{N}_0^+$,*

$$\frac{T_n\left(\sigma_2^\Lambda(\lambda)\right)}{T_n\left(\sigma_2^\Lambda(0)\right)} = \operatorname*{arg\,min}_{\substack{P \in \mathbb{R}_t[X], \\ P(0)=1}} \sup_{\lambda \in \Lambda} |P(\lambda)|, \quad \text{with } \sigma_2^\Lambda(\lambda) = 2\left(\frac{1+m}{2\sqrt{m}}\right)^2 \left(1 - \frac{\lambda}{L_1}\right)\left(1 - \frac{\lambda}{\mu_2}\right) - 1.$$

## 4.2 Generalization to Longer Cycles

The polynomial in Example 4.4 uses a linear link function $\sigma_1^\Lambda$ to map $\Lambda$ to $[-1, 1]$. In Proposition 4.5, we see that a degree *two* link function $\sigma_2^\Lambda$ can be used to find the minimax polynomial when $\Lambda$ is the union of two intervals. This section generalizes this approach and considers higher-order polynomials for $\sigma_K$. We start with the following parametrization, with an arbitrary polynomial $\sigma_K$ of degree $K$,

$$P_t(\lambda; \sigma_K) \triangleq \frac{T_n(\sigma_K(\lambda))}{T_n(\sigma_K(0))}, \quad \forall t = Kn, \, n \in \mathbb{N}_0^+ . \tag{14}$$

As we will see in the next subsection, this parametrization allows considering cycles of step-sizes. Our goal now is to find the $\sigma_K$ that obtains the fastest convergence rate possible. The next proposition quantifies its impact on the asymptotic rate and its proof can be found in Appendix D.1.

**Proposition 4.6.** *For a given $\sigma_K$ such that $\sup_{\lambda \in \Lambda} |\sigma_K(\lambda)| = 1$, the asymptotic rate factor $\tau^{\sigma_K}$ of the method associated to the polynomial* (14) *is*

$$1 - \tau^{\sigma_K} = \lim_{t \to \infty} \sqrt[t]{\sup_{\lambda \in \Lambda} |P_t(\lambda; \sigma_K)|} = \left( \sigma_0 - \sqrt{\sigma_0^2 - 1} \right)^{\frac{1}{K}}, \quad \text{with } \sigma_0 \triangleq \sigma_K(0) . \tag{15}$$

For a fixed $K$, the asymptotic rate (15) is a decreasing function of $\sigma_0$. This motivates the introduction of the "optimal" degree $K$ polynomial $\sigma_K^\Lambda$ as the one that solves

$$\sigma_K^\Lambda \triangleq \arg\max_{\sigma \in \mathbb{R}_K[X]} \sigma(0) \quad \text{s.t.} \quad \sup_{\lambda \in \Lambda} |\sigma(\lambda)| = 1 . \tag{16}$$

Using the above definition, we recover the $\sigma_1^\Lambda$ and $\sigma_2^\Lambda$ from Example 4.4 and Proposition 4.5.

**Finding the polynomial.** Finding an exact and explicit solution for the general $K$ and $\Lambda$ case is unfortunately out of reach, as it involves solving a potentially difficult system of $K$ non-linear equations. Here we describe an approximate approach. Let $\sigma_K^\Lambda(x) = \sum_{i=0}^K \sigma_i x^i$. We propose to discretize $\Lambda$ into $N$ different points $\{\lambda_j\}$, then solve the linear problem

$$\max_{\sigma_i} \sigma_0 \quad \text{s.t.} \quad -1 \leq \sum_{i=0}^K \sigma_i \lambda_j^i \leq 1, \quad \forall j = 1, \ldots, N . \tag{17}$$

To check the optimality, it suffices to verify that the polynomial $\sigma_K^\Lambda$ satisfies the *equioscillation* property (Definition 4.3), as depicted in Figure 3.

**Remark 4.7** (Relationship between optimal and minimax polynomials)**.** For later reference, we note that the optimal polynomial $\sigma_K^\Lambda$ is equivalent to finding a minimax polynomial on $\Lambda$ and to rescale it. More precisely, $\sigma_K^\Lambda$ is optimal if and only if $\sigma_K^\Lambda / \sigma_K^\Lambda(0)$ is minimax.

## 4.3 Cyclical Heavy Ball and (Non-)asymptotic Rates of Convergence

We now describe the link between $\sigma_K^\Lambda$ and Algorithm 3. Using the recurrence for Chebyshev polynomials of the first kind in (14), we have $\forall t = Kn, \, n \in \mathbb{N}_0^+$,

$$\frac{T_{n+1}(\sigma_K^\Lambda(\lambda))}{T_{n+1}(\sigma_K^\Lambda(0))} = 2\sigma_K^\Lambda(\lambda) \left[ \frac{T_n(\sigma_K^\Lambda(\lambda))}{T_n(\sigma_K^\Lambda(0))} \right] \underbrace{\left[ \frac{T_n(\sigma_K^\Lambda(0))}{T_{n+1}(\sigma_K^\Lambda(0))} \right]}_{=a_n} - \left[ \frac{T_{n-1}(\sigma_K^\Lambda(\lambda))}{T_{n-1}(\sigma_K^\Lambda(0))} \right] \underbrace{\left[ \frac{T_{n-1}(\sigma_K^\Lambda(0))}{T_{n+1}(\sigma_K^\Lambda(0))} \right]}_{=b_n} .$$

It still remains to find an algorithm associated with this polynomial. To obtain one in the form of Algorithm 1, one can use the stationary behavior of the recurrence. From [Scieur and Pedregosa, 2020], the coefficients $a_n$ and $b_n$ converge as $n \to \infty$ to their fixed-points $a_\infty$ and $b_\infty$. We therefore consider here an asymptotic polynomial $\bar{P}_t(\lambda; \sigma_K^\Lambda)$, whose recurrence satisfies

$$\bar{P}_t(\lambda; \sigma_K^\Lambda) = 2a_\infty \sigma_K^\Lambda(\lambda) \bar{P}_{t-K}(\lambda; \sigma_K^\Lambda) - b_\infty \bar{P}_{t-2K}(\lambda; \sigma_K^\Lambda) . \tag{18}$$

Similarly to $K = 1$, where this limit recursion corresponds to PHB, this recursion corresponds to an instance of Algorithm 3 (see Proposition 4.9 below), further motivating the cyclical heavy ball algorithm.

The following theorem is the main result of this section and characterizes the convergence rate of Algorithm 1 for arbitrary momentum and step-size sequences $\{h_i\}_{i \in [\![1,K]\!]}$. By optimizing over these parameters, we obtain a method associated to (18), whose rate is described in Proposition 4.9. All proofs can be found in Appendix D.2.

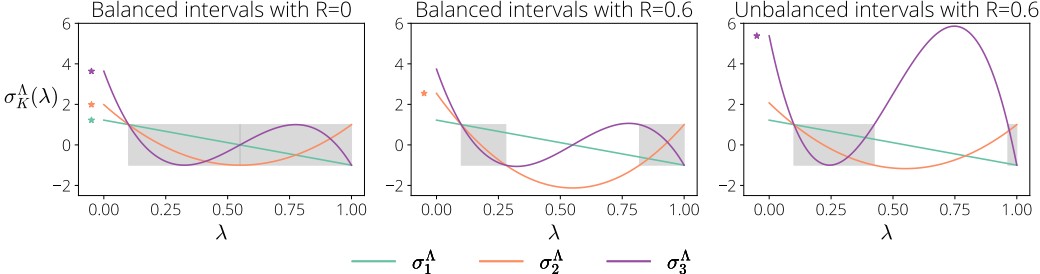

Figure 3: Examples of optimal polynomials $\sigma_K^\Lambda$ from (16), all of them verifying the equioscillation property (Definition 4.3). The "$\star$" symbol highlights the degree of $\sigma_K^\Lambda$ that achieves the best asymptotic rate $\tau^{\sigma_K^\Lambda}$ in (15) amongst all $K$ (see Section 4.4). **(Left)** When $\Lambda$ is an unique interval, all 3 polynomials are equivalently optimal $\tau^{\sigma_1^\Lambda} = \tau^{\sigma_2^\Lambda} = \tau^{\sigma_3^\Lambda}$. **(Center)** When $\Lambda$ is the union of two intervals of the same size, the degree 2 polynomial is optimal $\tau^{\sigma_2^\Lambda} > \tau^{\sigma_3^\Lambda} > \tau^{\sigma_1^\Lambda}$. This is expected given the result in Proposition 4.5. **(Right)** When $\Lambda$ is the union of two unbalanced intervals, the degree 3 polynomial instead achieves the best asymptotic rate $\tau^{\sigma_3^\Lambda} > \tau^{\sigma_2^\Lambda} > \tau^{\sigma_1^\Lambda}$ (see Section 4.4).

**Theorem 4.8.** *The worst-case rate of convergence of Algorithm 1 on $\mathcal{C}_\Lambda$ with an arbitrary momentum $m$ and an arbitrary sequence of step-sizes $\{h_i\}$ is*

$$1 - \tau = \begin{cases} \sqrt{m}, & \text{if } \sigma_{\text{sup}} \leq 1 \\ \sqrt{m}\left(\sigma_{\text{sup}} + \sqrt{\sigma_{\text{sup}}^2 - 1}\right)^{1/K}, & \text{if } \sigma_{\text{sup}} \in \left(1, \dfrac{1 + m^K}{2\left(\sqrt{m}\right)^K}\right) \\ \geq 1 \text{ (no convergence)} & \text{if } \sigma_{\text{sup}} \geq \dfrac{1 + m^K}{2\left(\sqrt{m}\right)^K} \end{cases} , \qquad (19)$$

*where $\sigma_{\text{sup}} \triangleq \sup\limits_{\lambda \in \Lambda} |\sigma(\lambda; \{h_i\}, m)|$, and $\sigma(\lambda; \{h_i\}, m)$ is the $K$-degree polynomial*

$$\sigma(\lambda; \{h_i\}, m) \triangleq \frac{1}{2}\text{Tr}\left(\begin{bmatrix} \frac{1+m-h_{K-1}\lambda}{\sqrt{m}} & -1 \\ 1 & 0 \end{bmatrix}\begin{bmatrix} \frac{1+m-h_{K-2}\lambda}{\sqrt{m}} & -1 \\ 1 & 0 \end{bmatrix} \cdots \begin{bmatrix} \frac{1+m-h_0\lambda}{\sqrt{m}} & -1 \\ 1 & 0 \end{bmatrix}\right). \quad (20)$$

**Proposition 4.9.** *Let $\sigma(\lambda; \{h_i\}, m)$ be the polynomial defined by (20), and $\sigma_K^\Lambda$ be the optimal link function of degree $K$ defined by (16). If the momentum $m$ and the sequence of step-sizes $\{h_i\}$ satisfy*

$$\sigma(\lambda; \{h_i\}, m) = \sigma_K^\Lambda(\lambda), \qquad (21)$$

*then **1)** the parameters are optimal, in the sense that they minimize the asymptotic rate factor from Theorem 4.8, **2)** the optimal momentum parameter is*

$$m = \left(\sigma_0 - \sqrt{\sigma_0^2 - 1}\right)^{2/K}, \quad \text{where } \sigma_0 = \sigma_K^\Lambda(0), \qquad (22)$$

***3)** the iterates from Algo. 3 with parameters $\{h_i\}$ and $m$ form a polynomial with recurrence (18), and **4)** Algorithm 3 achieves the worst-case rate $r_t^{\text{Alg. 3}}$ and the asymptotic rate factor $1 - \tau^{\text{Alg. 3}}$*

$$r_t^{\text{Alg. 3}} = O\left(t\left(\sigma_0 - \sqrt{\sigma_0^2 - 1}\right)^{t/K}\right), \qquad 1 - \tau^{\text{Alg. 3}} = \left(\sigma_0 - \sqrt{\sigma_0^2 - 1}\right)^{1/K}. \qquad (23)$$

**Solving the system (21)** The system is constructed by identification of the coefficients in both polynomials $\sigma_K^\Lambda$ and $\sigma(\lambda; \{h_i\}, m)$, which can be solved using a naive grid-search for instance. We are not aware of any efficient algorithm to solve this system exactly, although it is possible to use iterative methods such as steepest descent or Newton's method.

### 4.4 Best Achievables Worst-case Guarantees on $\mathcal{C}_\Lambda$

This section discusses the (asymptotic) optimality of Algorithm 3. In Section 4.2, the polynomial $P_t(\,\cdot\,;\sigma_K^\Lambda)$ was written as a composition of Chebyshev polynomials with $\sigma_K^\Lambda$, defined in (16). The best $K$ is chosen as follows: we solve (16) for several values of $K$, then pick the smallest $K$ among the minimizers of (15). However, following such steps does not guarantee that the polynomial $P_{t,K}^\Lambda$ is *minimax*, as it is not guaranteed to minimize the worst-case rate $\sup_{\lambda\in\Lambda}|P_t(\lambda)|$ (see (11)).

We give here an optimality certificate, linked to a generalized version of *equioscillation*. In short, if we can find $K$ non overlapping intervals (more formally, whose interiors are disjoint) $\Lambda_i$ in $\Lambda$ such that $\sigma_K^\Lambda(\Lambda_i) = [-1,1]$ then $P_{t,K}^\Lambda$ is minimax for all $t = nK$, $n \in \mathbb{N}_0^+$. The detailed result is provided by Theorem C.2. A direct consequence of this result is the asymptotic optimality of Algorithm 3, i.e., there exists no first order algorithm with a better asymptotic rate $1 - \tau$ for the function class $\mathcal{C}_\Lambda$.

It is possible that such $\sigma_K^\Lambda$ does not exist for a given $\Lambda$. A complete characterization of the set $\Lambda$ for which there exists such $\sigma_K^\Lambda$ is out of the scope of this paper. A partial answer is given in [Fischer, 2011] when $\Lambda$ is the union of two intervals. However, the problem remains open in the general case.

## 5 Local Convergence for Non-Quadratic Functions

When $f$ is twice-differentiable, it is possible to show local convergence rates when $x_0$ is close enough to $x_*$ [Polyak, 1964]. We give here a similar result that applies to Algorithm 1 (see proof in Appendix E). Those results are only local, as it is possible to find pathological counter-examples for which even PHB does not converge globally, for some specific initialization [Lessard et al., 2016].

**Theorem 5.1** (Local convergence). *Let $f : \mathbb{R}^d \mapsto \mathbb{R}$ be a (potentially non-quadratic) twice continuously differentiable function, $x_*$ a local minimizer, and $H$ be the Hessian of $f$ at $x_*$ with $Sp(H) \subseteq \Lambda$. Let $x_t$ denote the result of running Algorithm 1 with parameters $h_1, h_2, \cdots, h_K, m$, and let $1 - \tau$ be the linear convergence rate on the quadratic objective* (OPT). *Then we have*

$$\forall \varepsilon > 0, \exists \text{ open set } V_\varepsilon : x_0,\, x_* \in V_\varepsilon \implies \|x_t - x_*\| = O((1 - \tau + \varepsilon)^t)\|x_0 - x_*\|. \tag{24}$$

In short, when Algorithm 1 is guaranteed to converge at rate $1 - \tau$ on (OPT), then the convergence rate on a nonlinear functions can be arbitrary close to $1 - \tau$ when $x_0$ is sufficiently close to $x_*$.

## 6 Experiments

In this section we present an empirical comparison of the cyclical heavy ball method for different length cycles across 4 different problems. We consider two different problems, quadratic and logistic regression, each applied on two datasets, the MNIST handwritten digits [Le Cun et al., 2010] and a synthetic dataset. The results of these experiments, together with a histogram of the Hessian's eigenvalues are presented in Figure 4 (see caption for a discussion).

**Dataset description.** The MNIST dataset consists of a data matrix $A$ with 60000 images of handwritten digits each one with $28 \times 28 = 784$ pixels. The *synthetic* dataset is generated according to a spiked covariance model [Johnstone, 2001], which has been shown to be an accurate model of covariance matrices arising for instance in spectral clustering [Couillet and Benaych-Georges, 2016] and deep networks [Pennington and Worah, 2017, Granziol et al., 2020]. In this model, the data matrix $A = XZ$ is generated from a $m \times n$ random Gaussian matrix $X$ and an $m \times m$ deterministic matrix $Z$. In our case, we take $n = 1000, m = 1200$ and $Z$ is the identity where the first three entries are multiplied by 100 (this will lead to three outlier eigenvalues). We also generate an $n$-dimensional target vector $b$ as $b = Ax$ or $b = \text{sign}(Ax)$ for the quadratic and logistic problem respectively.

**Objective function** For each dataset, we consider a quadratic and a logistic regression problem, leading to 4 different problems. All problems are of the form $\min_{x\in\mathbb{R}^p} \frac{1}{n}\sum_{i=1}^n \ell(A_i^\top x, b_i) + \lambda\|x\|^2$, where $\ell$ is a quadratic or logistic loss, $A$ is the data matrix and $b$ are the target values. We set the regularization parameter to $\lambda = 10^{-3}\|A\|^2$. For logistic regression, since guarantees only hold at a neighborhood of the solution (even for the 1-cycle algorithm), we initialize the first iterate as the result of 100 iteration of gradient descent. In the case of logistic regression, the Hessian eigenvalues are computed at the optimum.

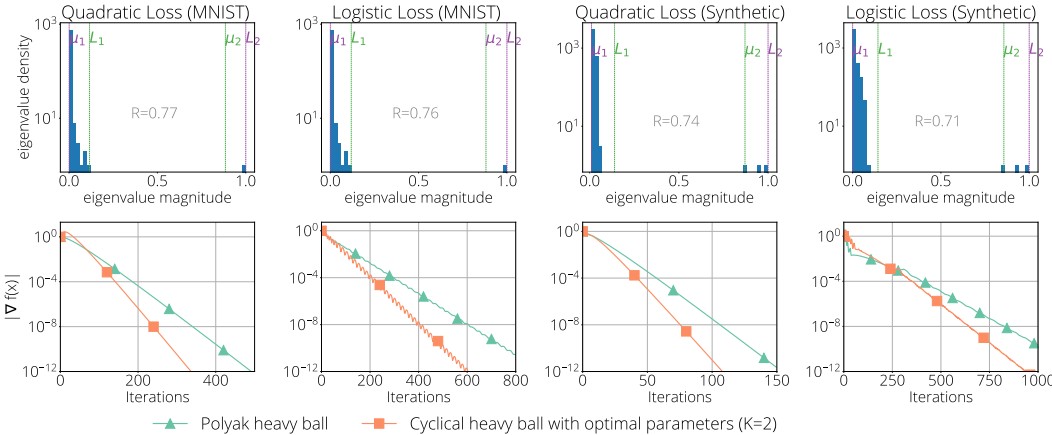

Figure 4: *Hessian Eigenvalue histogram (top row) and Benchmarks (bottom row)*. The **top row** shows the Hessian eigenvalue histogram at optimum for the 4 problems consider, together with the interval boundaries $\mu_1 < L_1 < \mu_2 < L_2$ for the two-interval split of the eigenvalue support described in Section 3. In all cases, there's a non-zero gap radius $R$. This is shown in the **bottom row**, where we compare the suboptimality in terms of gradient norm as a function of the number of iterations. As predicted by the theory, the non-zero gap radius translates into a faster convergence of the cyclical approach, compared to PHB in all cases. The improvement is observed on both quadratic and logistic regression problems, even through the theory for the latter is limited to *local* convergence.

## 7  Conclusion

This work is motivated by two recent observations from the optimization practice of machine learning. First, cyclical step-sizes have been shown to enjoy excellent empirical convergence [Loshchilov and Hutter, 2017, Smith, 2017]. Second, *spectral gaps* are pervasive in the Hessian spectrum of deep learning models [Sagun et al., 2017, Papyan, 2018, Ghorbani et al., 2019, Papyan, 2019]. Based on the simpler context of quadratic convex minimization, we develop a convergence-rate analysis and optimal parameters for the heavy ball method with cyclical step-sizes. This analysis highlights the regimes under which cyclical step-sizes have faster rates than classical accelerated methods. Finally, we illustrate these findings through numerical benchmarks.

**Main Limitations.**   In Section 3 we gave explicit formulas for the optimal parameters in the case of the 2-cycle heavy ball algorithm. These formulas depend not only on extremal eigenvalues—as is usual for accelerated methods—but also on the spectral gap $R$. The gap can sometimes be computed after computed the top eigenvalues (e.g. top-2 eigenvalue for MNIST). However, in general, there is no guarantee on how many eigenvalues are needed to estimate it. Moreover, global convergence result rely heavily on the quadratic assumption.

Another limitation regards long cycles. For cycles longer than 2, we have only given an implicit formula to set the optimal parameters (Proposition 4.9). This involves solving a set of non-linear equations whose complexity increases with the cycle length. That being said, cyclical step-sizes might significantly enhance convergence speeds both in terms of worst-case rates and empirically, and this work advocates that new tuning practices involving different cycle lengths might be relevant.

**Broader Impact.**   This work is mostly theoretical, and as such we believe it does not present direct societal consequences. However, the methods described in this paper can be used to train machine learning models which could themselves have societal consequences. For example, the deployment of machine learning models in decision-making has been shown to suffer from gender and racial bias and to amplify existing inequalities, see for instance [Hutchinson and Mitchell, 2019, Barocas et al., 2017, Obermeyer et al., 2019].

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
