# OpenReview forum: "Super-Acceleration with Cyclical Step-sizes"
_NeurIPS.cc/2021/Conference — NeurIPS 2021 Submitted_

### Official Review · Reviewer_a5zh · 2021-06-27

**Rating:** 5
**Confidence:** 4

**Summary:**

This paper studies using cyclical step-sizes of Heavy Ball for solving strongly convex quadratic problems. The authors first study setting the step size to two specific values periodically (period=2), where the values are related to the spectrum (i.e. the eigenvalues) of the underlying quadratic matrix. Assuming the spectrum can be divided into two groups. The authors prove that the resulting asymptotic rate of using the cyclical step size can be faster than that of the standard step size of Heavy Ball. Then, the authors discuss the extension to the case that the step size can have K values and the step size is set to the possible step sizes in a cyclical fashion.


**Limitations And Societal Impact:**

The authors adequately addressed the limitations and potential negative societal impact of their work.

**Main Review:**

I have some challenges for the authors.




=== 1. The paper missed a highly relevant reference: Samet Oymak. "Super-Convergence with an Unstable Learning Rate" arXiv:2102.10734 February. 2021.

The result of showing that if the spectrum of the Hessian can be grouped into two clusters, then using a cyclical Learning Rate can lead to a faster convergence compared to using the standard step size has been shown in [Oymak 2021]. Though I understand [Oymak 2021] is for gradient descent while this paper is for Heavy Ball, the novelty of this paper is not very high from this perspective.


=== 2. The intuition behind the acceleration result due to the cyclical step sizes seems not be explicitly described in the paper.


After reading the paper, I'm still not sure where exactly did the authors provide an explanation why the cyclical step sizes help faster convergence. It will be more helpful for the readers if the authors can be more transparent about their acceleration result.

On the other hand, [Oymak 2021] conveys a very clear message that periodically setting a large step size can help the progress on directions that correspond to small eigenvalues, while a small step size help to prevent divergence, and hence shows the benefit of using cyclical step sizes.


=== 3. The theoretical results are hard to interpret.



(3.a) The work makes an assumption that the eigenvalues are located in a way so that there are two intervals of the same size (line 61). It appears to be a strong assumption so the authors might want to discuss if it is an artifact of analysis.

(3.b) (Theorem 3.1) It is not easy to tell what would be the optimal momentum parameter $m$ and what would be the optimal step sizes of $h_1$ and $h_2$ from Theorem 3.1. Also, the choice of the momentum parameter $m$ and the step sizes $h_1$ and $h_2$ determines $\sigma_{\sup}$ in a seemingly complicated way. The authors might want to explain the interaction between these parameters. It would be better to present Theorem 3.1 in a more user-friendly fashion.

(3.c) (Section 4.2 and 4.3) The authors consider using cyclical step sizes with a longer cycle than 2 in the later parts of this paper. Does the result there show any benefit of using a longer cyclical step size? If so, can the authors provide a discussion here? If not, what would be the point of showing the result?




=== 4. Significance


The result in this paper seems to be very specific to the strongly convex quadratic problems, where the Hessian is a fixed matrix. If the Hessian changes a lot, then the analysis/techniques in this paper would likely be useless. So it is unclear if this work really makes an important step towards theoretically understanding the success of cyclical step sizes in practice.

A relevant question: Does the insight from the theoretical results in this paper leads to any new heuristic in optimization?


************* after rebuttal **************

I decided to leave the score as it is, because I think the idea has a great similarity to [Oymak 2021], and the proof techniques are not particularly novel and/or are standard for Heavy Ball (e.g. [Pedregosa 2021]). It is unclear to me if this paper makes another important step towards understanding cyclical step sizes in practice, compared with [Oymak 2021]. I don't think if the insights of this paper lead to any practical heuristics, and the practical contributions seem not central to the paper.

Also, the concern about the interaction of the parameters which was also raised by Reviewer 9ZCq is lingering.

Ref:
[Pedregosa 2021]
http://fa.bianp.net/blog/2021/hitchhiker/

**Time Spent Reviewing:**

6

---

> ### Author Response · Authors · 2021-08-10
> **Response to reviewer a5zh**
>
> We thank the reviewers for the time they spent on the review, their careful readings, and their encouraging comments. The paper will be revised accordingly; the current modifications also contain a series of smaller improvements to style and clarity.
>
> We would like to first clarify a few common points raised by the reviewers, before going into the detailed questions of each reviewer.
>
> ### Motivations
>
> As mentioned in the introduction, this work is motivated by two empirical observations: (i) the heavy ball method combined with cyclical step-sizes has been observed to improve convergence ; and (ii) it has been empirically observed that spectrum of the Hessians contains one or several gaps in their spectrum.
>
> One of our main contributions is to show that these observations are linked on quadratic functions, as cyclical step-sizes offer a simple way to exploit such gaps. As far as we know, such cyclical step-sizes are so far the only way to exploit such spectral properties efficiently.
>
> ### Quadratic optimization
> Motivated by second-order Taylor expansions and by simplicity, we focus in this paper on quadratic problems.
>     The guarantees/methods developed for quadratic minimization are usually transferred to more general setups either locally (as in Theorem 5.1 of the paper), or globally by using additional "globalization" strategies (see, e.g., "Numerical Optimization" by Nocedal and Wright).
>
> We admit however that there are important differences between our assumptions and those in the motivating work (stochastic vs deterministic, quadratics VS. nonlinear/neural networks).
>
> We added a paragraph to the Main limitations section precisely describing these differences. Nevertheless, we humbly believe that this work paves the way toward a better understanding of cyclical step-sizes by first analyzing the quadratic (or locally quadratic) case.
>
> Here are our answers to reviewer a5zh.
>
> -  (1. Oymak's preprint) We thank the reviewer for pointing us to this recent preprint (February 2021). We have added this reference to the current manuscript. Note however that there are significant differences between this and our work. As the reviewer kindly notes, Oymak's paper analyzes a gradient descent-like algorithms without momentum.
>
>     However, we respectfully disagree that this is a minor change and "novelty is not very high from this perspective". The introduction of momentum (i) makes the analysis much more challenging requiring a different proof technique, and (ii) allows to achieve faster rates than accelerated methods like Polyak momentum, while Oymak's method focuses on plain gradient descent.
>
>     As a side note, we recently had a very pleasant exchange with Samet Oymak, in which he pointed us towards his paper and underlined the differences between both works mentioned above.
>
> - (2. Missing intuition) While we agree the proof is rather technical, we have strived to expose intuitions underlying the geometrical properties of optimal polynomials (Figure 3), as well as the intuitive consequences of our analysis on the step-sizes (Figure 2).
>
>     Concerning an informal intuition for $K=2$, we believe it is similar in spirit to that for gradient descent with cyclical step-sizes. That is, the largest step-size helps converging faster in direction with low curvature, while smallest step-size helps converging faster in direction with high curvature. However, accounting for momentum, the potential existence of multiples clusters in the spectrum, and the possible larger values for $K$, to the best of our knowledge, advanced proof techniques are not based on this informal interpretation.
>
> - (3.a. Spectral assumptions: intervals of the same sizes) We believe there might be a misunderstanding as the assumption in that section is that the eigenvalues  _are contained_ in a union of intervals of the same size, not that the support is a union of intervals of the same size. It is possible to have intervals of different sizes and take as support set $[\mu_1,L_1']\cup[\mu_2',L_2]$ with $L_1'=\max\{L_1,\mu_1+ (L_2-\mu_2) \}$  and $\mu_2'=\min\{\mu_2,L_2- (L_1-\mu_1)\}$. In other words, take the largest of the two intervals, and make the other one of the same size. This still gives an accelerated method, although not the optimal one (the optimal one has cycles of length $K>2$ in this case).
>
>     Our paper also provides a way to design optimal methods for uneven intervals (if we allow $K>2$). See our answer to 3.c) for more details.
>
> - (3.b. Expression of Theorem 3.1 in a user-friendly fashion and computation of optimal parameters)
>
>     Due to the analysis, all the claims of this paper are tight. We made our best to present them in a human-readable way, but we currently do not believe they can be significantly simplified further.
>
>     Furthermore, although it might seem complicated to compute the optimal choices for $h_0, h_1$ and $m$ from Theorem 3.1, this is nevertheless achieved in closed form in Algorithm 2 and Corollary 3.2. We also summarized asymptotic behavior of this rate when $\kappa$ is close to 0 in Table 1, for clarity and interpretability.
>
> - (3.c. Longer cycles of step-sizes)  As far as the theory is concerned, the cases $K>2$ might be significant in various scenarios. For example, the reviewer raises the concern (question 3.a) that we assumed the eigenvalues are clustered within 2 intervals of the same size. As discussed above, reducing the size of one of the two intervals (i.e., using a stronger assumptions) does no lead to faster convergence (than the case where the two intervals have the same size) using only $K=2$. Yet in this case, there are strategies with *longer cycles* that can fully take advantage of the structure of eigenvalues; the precise cycle length depending on the particular values of the intervals. We provide an example with $K=3$ in Appendix D4.
>
> - (4. Beyond deterministic quadratic minimization) Many (probably most) theoretical contributions in first-order optimization were initially discovered and developed for/on quadratics. Some of those contributions never even went past that stage, including common practical ones (for example, heavy ball does not improve worst-case convergence bounds beyond quadratics). This is a first step towards the understanding of the behavior of first-order methods based on cyclical step-sizes strategies, and we believe it is reasonable to leave the next steps for future work.
>
> - (5. New heuristic?) As mentioned in the introduction, our goal is to justify the use of cyclical step-sizes --a common heuristic for accelerating convergence-- within momentum-based methods in a simple setting rather than to propose new heuristics.
>
>     Being aware of working parameterization of such methods could potentially allow designing new methods (not only heuristics). For example, heavy ball was already used in the context of quadratic minimization. For HB users, performing an additional (logarithmic) grid search over $R$ (the gap in the spectrum) potentially allows obtaining nearly optimal complexities in this setup, just as it is common to do for estimating the strong convexity constants (see, e.g., "Sharpness, Restart and Acceleration" (2020) by V. Roulet and A. d'Aspremont).
>
> We would like to thank one more time all the reviewers for their time and constructive comments on our paper.
> Please feel free to ask for any other clarifications.

---

> > ### Comment · Reviewer_a5zh · 2021-08-14
> > **comment**
> >
> > I thank the authors for their response.
> >
> > But I don't think the authors have clarified my concern (3.b). The dependency of parameters seems complicated. Suppose a user just wants to set the momentum parameter to be $(1-c \sqrt{1/\kappa})^2$, where $c>0$ is a universal constant, and $\kappa$ is the condition number. How to choose the other parameters optimally? What is the convergence rate? What if the user sets the momentum parameter to be 0.9? I don't think it is easy to answer these questions from Theorem 3.1. So it will be more helpful if the authors can give some guidance on how to use their results.
> >
> > Furthermore, the convergence rates are asymptotic. The rates are about the case when iteration $t \rightarrow \infty$. It is unclear if the theoretical results truthfully capture what goes on in a finite $t$. On the other hand, a non-asymptotic rate of Heavy ball does exist, e.g. Section 3.2 in https://arxiv.org/pdf/1901.07445.pdf

---

> > > ### Author Response · Authors · 2021-08-16
> > > **Response to reviewer a5zh about Theorem 3.1**
> > >
> > > We thank the reviewer a5zh for having carefully read our answers. We are pleased that almost all their concerns about the paper are clarified, and in the meantime sorry to learn that one is not clarified yet.
> > >
> > > - In the case where $m$ is fixed to an arbitrary value, we agree that finding the optimal values analytically for $h_0$ and $h_1$ from Theorem 3.1 is not straightforward. However, for each value of $(h_0, h_1)$, Theorem 3.1 provides the associated rate, which can be computed numerically. For optimizing the rate, it is then possible to do a 2D grid search over the values of the step-sizes. Please also note that for a fixed $m$, the rate is an increasing function of $\sigma_{\sup}$ (which is the max of 5 values, depending on $h_0$ and $h_1$). Therefore, one only needs to compute the latter at each point of the grid (because $\sigma_{\sup}$ is a function of $h_0$ and $h_1$).
> > > - We decided to present asymptotic convergence rates striving to make the convergence rate results as simple as possible, but it is easy to derive non-asymptotic results from the existing proofs. More precisely, we can derive non-asymptotic rates for the case $K=2$ following the proof of Theorem 4.8 in Appendix D.2, starting from eq. (134), and using standard bounds of Chebyshev polynomials. We would gladly add non-asymptotic convergence rate results if the reviewer believes this would improve the paper. Please also note that Polyak Heavy-Ball is asymptotically worst-case optimal, and corresponds to the stationary version of the so-called "Chebyshev semi-iterative method", which is the ultimately worst-case optimal method for smooth strongly convex quadratic minimization. In the same vein, our algorithm is asymptotically optimal, and the corresponding "totally optimal method" is presented in Appendix D.1 (See Algorithm 6). It corresponds to Chebyshev semi-iterative method when R (the gap) is exactly 0.
> > >
> > > We hope this clarifies your last concern. If not, please feel free to let us know.

---

### Official Review · Reviewer_ao6p · 2021-07-06

**Rating:** 6
**Confidence:** 4

**Summary:**

This paper studies cyclical step sizes for heavy ball momentum. It is shown that under some conditions, it is possible to improve the convergence rate of heavy ball by relying on cyclical step sizes with K = 2. The result is obtained by a careful examine of the spectrum of the Hessian. The cases for larger K are also discussed. In general, the results are nice but also has its own limitation.

**Limitations And Societal Impact:**

This is mainly a theoretical work. The Societal impact depends on how it is applied to solve real problems.

**Main Review:**

Strength:

(+) This work demonstrates that cyclic step sizes are useful and theoretically improves over HB under certain conditions. I believe the results are of interests in the community.

(+) The paper is well written and easy to follow.


Weakness

(-) Though theoretically appealing, it is unclear that the improvement comes from. In particular, is it possible to tighten the HB convergence as well assuming (3) is true? This is a critical question since without demonstrating it, it is not possible to conclude cyclic step sizes are helpful.

(-) For quadratic problems, it is relative easy to find L2 and mu2, but is it common in e.g., logistic regression or other loss functions to find these parameters?

**Time Spent Reviewing:**

2

---

> ### Author Response · Authors · 2021-08-10
> **Response to reviewer ao6p**
>
> We thank the reviewers for the time they spent on the review, their careful readings, and their encouraging comments. The paper will be revised accordingly; the current modifications also contain a series of smaller improvements to style and clarity.
>
> We would like to first clarify a few common points raised by the reviewers, before going into the detailed questions of each reviewer.
>
> ### Motivations
>
> As mentioned in the introduction, this work is motivated by two empirical observations: (i) the heavy ball method combined with cyclical step-sizes has been observed to improve convergence ; and (ii) it has been empirically observed that spectrum of the Hessians contains one or several gaps in their spectrum.
>
> One of our main contributions is to show that these observations are linked on quadratic functions, as cyclical step-sizes offer a simple way to exploit such gaps. As far as we know, such cyclical step-sizes are so far the only way to exploit such spectral properties efficiently.
>
> ### Quadratic optimization
> Motivated by second-order Taylor expansions and by simplicity, we focus in this paper on quadratic problems.
>     The guarantees/methods developed for quadratic minimization are usually transferred to more general setups either locally (as in Theorem 5.1 of the paper), or globally by using additional "globalization" strategies (see, e.g., "Numerical Optimization" by Nocedal and Wright).
>
> We admit however that there are important differences between our assumptions and those in the motivating work (stochastic vs deterministic, quadratics VS. nonlinear/neural networks).
>
> We added a paragraph to the Main limitations section precisely describing these differences. Nevertheless, we humbly believe that this work paves the way toward a better understanding of cyclical step-sizes by first analyzing the quadratic (or locally quadratic) case.
>
> Please find below our answers to reviewer ao6p.
>
> - (1. Improving the analysis of heavy ball under spectral assumptions) One possible interpretation of what happens is that the largest step-size helps converging faster in directions of small curvature, while smallest step-size helps converging faster in directions of higher curvature.
>
>     More rigorously, without the additional spectral gap assumption, the classical heavy ball tuning with constant step-size is optimal (among black-box first-order methods). With the additional spectral gap assumption, the worst-case performance of the classical heavy ball is not improved (because heavy ball achieves its worst-case, among others, on the boundary of the spectrum; i.e., $L x^2/2$ and $\mu x^2/2$ are among the worst-case functions), showing that cyclical step-sizes indeed improves over heavy ball in this context.
>
> - (2. How to estimate $L_2$, $\mu_2$) We are not totally sure whether the reviewer really meant "$L_2$ and $\mu_2$" or another couple of eigenvalues, as L2 is the smoothness constant, and not the hardest to access. We interpreted the question with $L_1$ and $\mu_2$ and answered accordingly. Please let us know if we misinterpreted this point.
>
>     We do not believe $L_1$ and $\mu_2$ are so easy to compute, even for quadratics (computing the spectrum would probably not be a very reasonable option in most large-scale problems).
>
>     We agree that computing the algorithmic parameters requires knowledge of additional problem parameters and would like to note that we do not claim that estimating them is always practical. However, in some cases they can be estimated using a top-k eigenvalue distribution. For the problems considered, k = 3 has been sufficient, although the method can be inefficient for large k. For the problems we considered, performing a top-3 eigenvalue decomposition using the iterative solver ARPACK was faster than computing the smallest eigenvalue. Another way to estimate the parameters is through a grid-search over the gap, adding a constant factor over the overall complexity.
>
>     Furthermore, we would like to emphasize that our main contribution is to develop the theoretical understanding of the role of cyclical strategies, and not necessarily to provide a practical way of tuning the algorithm parameters in all situations.
>
>     Developing adaptive methods that don't require knowledge of such problem constants is also a point of interest for the community, and it remains in general open even in simpler constant step-size settings. For instance, to the best of our knowledge there is no "natural" accelerated method, even in the smooth strongly convex minimization setting, being adaptive to the strong convexity parameter that does not require restart strategies.
>
> We would like to thank one more time all the reviewers for their time and constructive comments on our paper.
> Please feel free to ask for any other clarifications.

---

### Official Review · Reviewer_HKjc · 2021-07-11

**Rating:** 6
**Confidence:** 4

**Summary:**

This paper studies the cyclicial step-sizes to improve the worst-case convergence of the heavy ball method (deterministic version). More precisely, the authors 1) achieve a faster asymptotic worst-case convergence  rate compared to the polayk heavy ball method for the strongly convex and L-smooth quadratic objectives,  2) show how to select the optimal momentum and step-size cycles, and 3) extend the local convergence to the non-quadratic case. Furthemore, numerical experiments on quadratics and logistic regressions have been conducted to demonstrate the main strength of the cyclicial heavy ball method.

**Limitations And Societal Impact:**

In general, the authors have adequately addressed the limitations and potential negative societal impact of their work.

**Main Review:**

Overall, the paper is well-written and the proof is solid. The novelty of  this paper is that the authors take advantage of the cyclical step-size strategies which recently have been widely used in the stochastic optimization setting, to improve the worst-case convergence of the traditional heavy ball method. The strong points of this paper are its theoretical parts: 1) the cyclicial heavy ball  for example with 2 cycles has a faster worst-case convergence over the traditional polyak heavy ball method (see Table 1); 2) provide a systematic approach to select the optimal parameters for momentum and step-size cycles if given the spectrum interval of Hessian $\Lambda$ and cycle length $K$.  I like the results, however 1) their analysis heavily relies on the distribution of the Hessian's eigenvalues which makes it uneasy to be extended to other general settings; 2) the algorithm needs to estimate some extra eigenvalues which may be expensive for large scale problems. I have the following detailed concerns or comments on this paper:

1) From Figure 2, we can see that when the eigenvalue gap $R \neq 0$, you have two choice for the optimal step-sizes $(h_0, h_1)$: 1) $h_0 > h_1$ and 2) $h_0 < h_1$. In Algorithm 2, the authors use the first choice that $h_0 > h_1$. If you choose another one i.e., $h_0 < h_1$,  could you get similar results? How will it affect your numerical results?
2) The cyclicial step-sizes indeed have been widely used in deep learning. However, it is not very clear why the cyclicial step-size strategy proposed in this paper helps to improve the traditional heavy ball method. Could you try to explain it a little bit?
3) The theoretical results for the cyclicial heavy ball method heavily rely on the location of the Hessian’s eigenvalues and also the quadratic assumption. Is it possible or easy to extend this idea to e.g., the general convex problems and nonconvex problems, especially in deep learning, or stochastic heavy ball method?
4) In Figure 4, from the first and the last pictures of the bottom row, we can see that the cyclicial heavy ball with optimal parameters performs worse than the polyak heavy ball method at the beginning. Could you clarify why this happens?
5) The computational cost: except the smallest ($\mu$) and largest ($L$) eigenvalues as polyak heavy ball, the proposed algorithm also needs to know the distribution of the eigenvalues, 1) if the eigenvalues of Hessian are clustered (the best situation) around the smallest or largest eigenvalues for example MNIST dataset (see the first picture of Figure 4), we still need to estimate other two top eigenvalues except the extremal eigenvalues (i.e., $\mu$ and $L$). So it will be costly and impractical for the large scale optimization problems. Especially, for the datasets in which the distribution of the Hessian’s eigenvalues is dense (not clustered), the computational cost will be highly increased but the improvements seem to be less because in this case the eigenvalue gap $R$ is smaller.  Could you summary in what kinds of situations the increasing of the computational cost can be ignored compared to its improvements.

Other Comments:
1) There are some mistakes in Table 1: 1) in the first line ($R \in [0,1)$), the value  of "Set $\Lambda$ " is wrong, it should be $[\mu, \mu + (L-\mu)(1-R)/2] \cup [L-(L-\mu)(1-R)/2, L]$;  2) in the third line and the fourth column($R=1-2\gamma \kappa$), the speedup should be “ $O(\kappa^{-1/2})$ ” instead of “$O(\kappa^{1/2})$”.
2) In the theoretical part, the authors use $m$ and $\lambda$, which appear frequently, to denote the momentum parameter and the eigenvalue of Hessian, respectively. But in the numerical part, $m$ is also used to denote the dimension of the data matrix and $\lambda$ is used to denote the regularization parameter. This easily leads to misunderstandings to the readers.
3) There is an additional "with" in the name of Algorithm 2.



**Time Spent Reviewing:**

18 hours

---

> ### Author Response · Authors · 2021-08-10
> **Response to reviewer HKjc**
>
> We thank the reviewers for the time they spent on the review, their careful readings, and their encouraging comments. The paper will be revised accordingly; the current modifications also contain a series of smaller improvements to style and clarity.
>
> We would like to first clarify a few common points raised by the reviewers, before going into the detailed questions of each reviewer.
>
> ### Motivations
>
> As mentioned in the introduction, this work is motivated by two empirical observations: (i) the heavy ball method combined with cyclical step-sizes has been observed to improve convergence ; and (ii) it has been empirically observed that spectrum of the Hessians contains one or several gaps in their spectrum.
>
> One of our main contributions is to show that these observations are linked on quadratic functions, as cyclical step-sizes offer a simple way to exploit such gaps. As far as we know, such cyclical step-sizes are so far the only way to exploit such spectral properties efficiently.
>
> ### Quadratic optimization
> Motivated by second-order Taylor expansions and by simplicity, we focus in this paper on quadratic problems.
>     The guarantees/methods developed for quadratic minimization are usually transferred to more general setups either locally (as in Theorem 5.1 of the paper), or globally by using additional "globalization" strategies (see, e.g., "Numerical Optimization" by Nocedal and Wright).
>
> We admit however that there are important differences between our assumptions and those in the motivating work (stochastic vs deterministic, quadratics VS. nonlinear/neural networks).
>
> We added a paragraph to the Main limitations section precisely describing these differences. Nevertheless, we humbly believe that this work paves the way toward a better understanding of cyclical step-sizes by first analyzing the quadratic (or locally quadratic) case.
>
> Find below our answers to reviewer HKjc.
>
> - (1. Step-sizes ordering) This is a very good question, which we clarify in the next version of the paper. The choice $h_0 > h_1$ in Algorithm 2 is arbitrary and it does not affect the results.
>     There are several ways to understand this phenomenon. In particular, one can notice that the polynomial analysis in Appendix D.3 is symmetric in $(h_0, h_1)$, and the convergence rates are stated for steps multiple of $K$. In the case $K=3$ all step-size permutations also work.
>     In the general case (any $K>0$), the polynomial analysis is invariant to any *cyclic permutation* of the sequence $(h_i)$.
>
> - (2. Improvement of traditional heavy ball method) One possible interpretation of what happens is that the largest step-size helps converging faster in directions of small curvature, while smallest step-size helps converging faster in directions of higher curvature.
>
>     More rigorously, without the additional spectral gap assumption, the classical heavy ball tuning with constant step-size is optimal (among black-box first-order methods). With the additional spectral gap assumption, the worst-case performance of the classical heavy ball is not improved (because heavy ball achieves its worst-case, among others, on the boundary of the spectrum; i.e., $L x^2/2$ and $\mu x^2/2$ are among the worst-case functions), showing that cyclical step-sizes indeed improves over heavy ball in this context.
>
> - (3. Beyond deterministic quadratic minimization) To the best of our knowledge, the proofs unfortunately do not directly translate beyond quadratics, nor to a stochastic setting. These extensions are left for future work.
>
>     That being said, we do provide experiments on non-quadratic objectives that show that cyclical strategies work well in practice on these problems. Furthermore, in "Acceleration via Fractal Learning Rate Schedules", Naman Agarwal, Surbhi Goel and Cyril Zhang proved a convergence result for the Gradient Descent algorithm in the additive noise model (See Eq. 2) by using a polynomial analysis. This leaves hope for potentially reusing similar proof techniques and for adapting our work to a more general settings.
>
> - (4. Experiments, transition phase) The algorithms were developed to have (black-box) optimal asymptotic convergence rates, so we expect the error to decrease faster with our method compared to standard competitors, such as heavy ball. However, this does not mean that the algorithms are optimal for any problem instance and any number of iterations. In other words, this does not exclude heavy ball to be better on some specific problems in the class for some number of iterations: algorithms might get lucky (or unlucky), depending on the problem instance. This is particularly true for early iterations, for which the asymptotic worst-case guarantee of cyclical step-sizes does not apply yet.
>
> - (5. Knowledge of additional spectral information) We agree that computing the algorithmic parameters requires knowledge of additional problem parameters and would like to note that we do not claim that estimating them is always practical. However, in some cases they can be estimated using a top-k eigenvalue distribution. For the problems considered, k = 3 has been sufficient, although the method can be inefficient for large k. For the problems we considered, performing a top-3 eigenvalue decomposition using the iterative solver ARPACK was faster than computing the smallest eigenvalue. Another way to estimate the parameters is through a grid-search over the gap, adding a constant factor over the overall complexity.
>
>     Furthermore, we would like to emphasize that our main contribution is to develop the theoretical understanding of the role of cyclical strategies, and not necessarily to provide a practical way of tuning the algorithm parameters in all situations.
>
>     Developing adaptive methods that don't require knowledge of such problem constants is also a point of interest for the community, and it remains in general open even in simpler constant step-size settings. For instance, to the best of our knowledge there is no "natural" accelerated method, even in the smooth strongly convex minimization setting, being adaptive to the strong convexity parameter that does not require restart strategies.
>
> - (Other comments) We thank the reviewer very much for raising these valid mistakes.
>
> We would like to thank one more time all the reviewers for their time and constructive comments on our paper.
> Please feel free to ask for any other clarifications.

---

> > ### Comment · Reviewer_HKjc · 2021-08-30
> > **Thanks for the response**
> >
> > Thanks for the response. I have carefully read the response and other reviews.

---

### Official Review · Reviewer_9ZCq · 2021-07-13

**Rating:** 6
**Confidence:** 4

**Summary:**

The authors study a variant of the heavy ball method, where the stepsizes are chosen in a cyclical order. The paper considers the problem of minimising quadratic functions (there is a paragraph on local analysis for more general functions, but this is reduced to  the quadratic case  by linearisation). In the case where the cycle is 2 and the intervals of the eigenvalues of the quadratic functions are known, the authors show how this cyclic stepsize choice can lead to acceleration. In the more general setting of cycle $K$, their analysis is subject to some polynomial.




**Limitations And Societal Impact:**

Yes

**Main Review:**


Overall, I found this paper interesting, and the theoretical study of good stepsize choices is important for stochastic algorithms.
However, in general, I'm not sure how practical/useful the results here actually are, Algorithm 3 requires finding a polynomial which may not exist and involves solving a difficult system of $K$ nonlinear equations. The $K=2$ scheme also requires not just estimating the largest and smallest eigenvalue, but also the eigenvalue gap. This is in contrast to  existing analysis into cyclical stepsizes in descent methods, see last paragraph.

I'm also a  confused about the statement in Theorem 3.1, it refers to Algorithm 1 where $h_0$ and $h_1$ are given explicitly in terms of the eigenvalue bounds and $m$, but you mention immediately after this theorem about optimising $h_0$ and $h_1$, so are these values fixed as in the Algorithm or are they arbitrary? If they are arbitrary, what the corresponding optimal $h_0$ and $h_1$ are which give the optimal rate in  corollary 3.2 (otherwise, after estimating the eigenvalue bounds, would one need to apply something like Newton's method to optimise for these step sizes? This could be potentially more costly than the acceleration obtained?) ?
Finally,  another question I have is: the motivation given in the introduction is from the use of restarts in stochastic algorithms, while the analysis here is in the deterministic setting, do the authors expect the results here to apply also in the stochastic? Perhaps some numerical illustrations on this would be helpful?

There is  a related strand of work on the use of cyclical stepsize for descent methods (with  many results appearing in the later 1990's and early 2000's)  -- the analysis into acceleration in  convergence rates  has also been studied in many works. These works make connections to the relationship between eigenvalues of the Hessian (similar to the analysis in this work), and also provide guidance on practical choices of stepsizes based on past iterates. Some examples: Friedlander et al, SIAM Journal on Numerical Analysis, Vol. 36, No. 1 (1999), pp. 275-289; Dai and Fletcher, Math. Program., Ser. A 103, 541–559 (2005); Dai et al, IMA Journal of Numerical Analysis (2006) doi:10.1093/imanum/drl006. See also the more recent work by Kalousek, Found Comput Math (2017) 17:359–42, DOI 10.1007/s10208-015-9290-8. Since this body of work very close to the results presented, the authors should  reference appropriately  and provide some discussions on the relation to their work.

**Time Spent Reviewing:**

4

---

> ### Author Response · Authors · 2021-08-10
> **Response to reviewer 9ZCq**
>
> We thank the reviewers for the time they spent on the review, their careful readings, and their encouraging comments. The paper will be revised accordingly; the current modifications also contain a series of smaller improvements to style and clarity.
>
> We would like to first clarify a few common points raised by the reviewers, before going into the detailed questions of each reviewer.
>
> ### Motivations
>
> As mentioned in the introduction, this work is motivated by two empirical observations: (i) the heavy ball method combined with cyclical step-sizes has been observed to improve convergence ; and (ii) it has been empirically observed that spectrum of the Hessians contains one or several gaps in their spectrum.
>
> One of our main contributions is to show that these observations are linked on quadratic functions, as cyclical step-sizes offer a simple way to exploit such gaps. As far as we know, such cyclical step-sizes are so far the only way to exploit such spectral properties efficiently.
>
> ### Quadratic optimization
> Motivated by second-order Taylor expansions and by simplicity, we focus in this paper on quadratic problems.
>     The guarantees/methods developed for quadratic minimization are usually transferred to more general setups either locally (as in Theorem 5.1 of the paper), or globally by using additional "globalization" strategies (see, e.g., "Numerical Optimization" by Nocedal and Wright).
>
> We admit however that there are important differences between our assumptions and those in the motivating work (stochastic vs deterministic, quadratics VS. nonlinear/neural networks).
>
> We added a paragraph to the Main limitations section precisely describing these differences. Nevertheless, we humbly believe that this work paves the way toward a better understanding of cyclical step-sizes by first analyzing the quadratic (or locally quadratic) case.
>
> Find below our answers to the points raised by reviewer 9ZCq.
>
> - (1. Beyond deterministic quadratic minimization) To the best of our knowledge, the proofs unfortunately do not directly translate beyond quadratics, nor to a stochastic setting. These extensions are left for future work.
>
>     That being said, we do provide experiments on non-quadratic objectives that show that cyclical strategies work well in practice on these problems. Furthermore, in "Acceleration via Fractal Learning Rate Schedules", Naman Agarwal, Surbhi Goel and Cyril Zhang proved a convergence result for the Gradient Descent algorithm in the additive noise model (See Eq. 2) by using a polynomial analysis. This leaves hope for potentially reusing similar proof techniques and for adapting our work to a more general settings.
>
> - (2. Knowledge of additional spectral information) We agree that computing the algorithmic parameters requires knowledge of additional problem parameters and would like to note that we do not claim that estimating them is always practical. However, in some cases they can be estimated using a top-k eigenvalue distribution. For the problems considered, k = 3 has been sufficient, although the method can be inefficient for large k. For the problems we considered, performing a top-3 eigenvalue decomposition using the iterative solver ARPACK was faster than computing the smallest eigenvalue. Another way to estimate the parameters is through a grid-search over the gap, adding a constant factor over the overall complexity.
>
>     Furthermore, we would like to emphasize that our main contribution is to develop the theoretical understanding of the role of cyclical strategies, and not necessarily to provide a practical way of tuning the algorithm parameters in all situations.
>
>     Developing adaptive methods that don't require knowledge of such problem constants is also a point of interest for the community, and it remains in general open even in simpler constant step-size settings. For instance, to the best of our knowledge there is no "natural" accelerated method, even in the smooth strongly convex minimization setting, being adaptive to the strong convexity parameter that does not require restart strategies.
>
> - (3. Computing the step-sizes) As the reviewer correctly pointed out, computing the optimal step-sizes for cycles longer than 2 requires solving a nonlinear system. This might indeed be an issue if this strategy is used for choosing step-sizes in practical applications.
>
>     That being said, one could note that (i) this system only needs to be solved once for all, that (ii) that we typically do not want to solve this system for large values of $K$, and that (iii) our theorems (Theorem 4.8 or Theorem 3.1 for $K=2$) provide convergence guarantees for all values of the parameters, and we can therefore reasonably solve the nonlinear system only approximately.
>
> - (4. Optimal tuning) As pointed out by the reviewer, Theorem 3.1 holds for Algorithm 1 (with $K=2$) for all values of $m$, $h_0$ and $h_1$.  Algorithm 2 is an instance of Algorithm 1 with the optimal parameters used in Corollary 3.2.
>
> - (5. References) We thank the reviewer for providing us those references. Those papers are indeed very interesting, but we would like to note that they mostly deal with a "cyclical" variant of Barzilai-Borwein method where the term cyclical here takes a different meaning. Indeed, they use the same step-size for a certain number of iterations before updating it (see, e.g., eq.3.3 of "Gradient Method with Retards and Generalizations", eq.1.8 of "On the asymptotic behaviour of some new gradient methods" or eq.1.7 of "The cyclic Barzilai--Borwein method for unconstrained optimization"). This is significantly different to the meaning we gave to "cyclical" in our paper.
>
> We would like to thank one more time all the reviewers for their time and constructive comments on our paper.
> Please feel free to ask for any other clarifications.

---

### Decision · Program_Chairs · 2021-09-27

**Decision:**

Reject

**Comment:**

Given the reviewers comments and after thorough discussion, it has been decided that this paper will not be accepted at NeurIPS 2021. The relationship to previous work and comments about parameter selection should be taken into consideration by authors when considering revisions of this manuscript.